# Risk Stratification of Patients with Acute Coronary Syndrome

**DOI:** 10.3390/jcm10194574

**Published:** 2021-10-01

**Authors:** Dávid Bauer, Petr Toušek

**Affiliations:** Department of Cardiology, Third Faculty of Medicine, Charles University, University Hospital Královské Vinohrady, 100 34 Prague, Czech Republic; petr.tousek@fnkv.cz

**Keywords:** acute coronary syndrome, percutaneous coronary intervention, risk stratification, risk score, prognosis

## Abstract

Defining the risk factors affecting the prognosis of patients with acute coronary syndrome (ACS) has been a challenge. Many individual biomarkers and risk scores that predict outcomes during different periods following ACS have been proposed. This review evaluates known outcome predictors supported by clinical data in light of the development of new treatment strategies for ACS patients during the last three decades.

## 1. Introduction

Prognosis of patients with acute coronary syndrome (ACS) has improved significantly in recent decades. However, there are continuous efforts to predict outcome of patients with ACS. Many biomarkers, scoring systems, and angiographic findings show promising results.

Based on clinical data, we would like to present a complex view on risk stratification methods. Focusing on scoring systems and anatomical features, we describe diagnostic and therapeutic progress of recent decades. Considering current guidelines, we make an up-to-date comparison of risk-predicting scoring systems for patients with ACS.

Our goal is to identify the most important variables for future stratification. We believe a simple, universal scoring system should have a role in everyday clinical practice.

## 2. Biomarkers

Several biomarkers have been proposed to define the prognosis of patients with ACS, many of which are routinely used in everyday clinical practice.

It has been suggested that an elevated N-terminal fraction of brain natriuretic peptide (NT-proBNP) level may be caused by transient myocardial ischemia, even without left ventricular systolic dysfunction [1]. An elevated NT-proBNP level in a patient with ACS successfully predicts a worse prognosis (mortality, development of left ventricular systolic dysfunction) [2]. Current European Society of Cardiology (ESC) guidelines recommend the use of NT-proBNP as a prognostic factor in patients with ACS (Class IIa, level B), but the use of other biomarkers is not recommended. [3].

High troponin levels predict an increased risk of mortality, and high-sensitivity troponin (hs-Tn)-T seems to be a better predictor than hs-Tn-I [4,5]. Current ESC guidelines recommend that troponin should be measured serially to determine the prognosis in patients with ACS (Class I, level B) [3].

However, the prognostic value is attenuated in older patients. Low troponin levels are associated more with high mortality in older patients with ACS than with younger patients. Physiological changes in older patients may be responsible for such results. [6]. Troponin is considered an independent risk factor for all-cause mortality in the general population (without ACS). Patients admitted to the emergency department for suspected ACS and who were subsequently discharged without an acute cardiovascular event have higher mortality in association with mildly elevated troponin (after 2 and 5.8 years of follow-up) [7,8].

Interestingly, higher troponin levels do not necessarily equal a higher risk of re-infarction, although mortality remains high [9].

The role of acute-phase proteins has been investigated in recent years. The concept that C-reactive protein (CRP) is produced solely by hepatic cells during inflammation has been questioned. Measuring CRP levels by real-time polymerase chain reaction demonstrate increased production due to unstable atherosclerotic plaques and their direct effect on endothelial dysfunction [10].

Animal studies show a negative effect on the myocardium after injecting human CRP into the rat coronary artery [11]. Several studies demonstrate a higher rate of adverse events and 30-day mortality in association with elevated CRP levels. [12].

CRP in combination with troponin adds prognostic value. Patients with a low CRP concentration (<10 mg/L) and a negative Tn-T or I have very low mortality rates (as low as 0% for up to 4 years). Higher mortality is expected if a high CRP concentration is present in a troponin-negative case of ACS (>2%) [12].

Atherosclerosis is a complex process in which the leukocyte-derived enzyme myeloperoxidase (MPO) plays an important role. Atherosclerosis involves modification of low-density lipoprotein particles, impaired endothelial function, destabilization of atherosclerotic plaques, and the induction of apoptosis [13]. The prognostic potential of MPO was first described Brennan in 2003 as an independent predictor of MACEs (myocardial infarction, re-infarction, the need for revascularization, or death) in the 6 months following admission to the emergency department for chest pain [14]. The CAPTURE trial enrolled 1265 patients with ACS. Higher MPO levels have significant event rates at 72-h, 30-day, and 6-month follow-up visits. Other studies confirm the potential of MPO as an independent prognostic factor in patients with ACS [15,16,17].

Apoptosis observed at the molecular level using the soluble apoptotic marker tumor necrosis factor-related apoptosis-stimulating ligand (TRAIL) is closely correlated with the prognosis of patients with ST elevation acute coronary syndrome (STE-ACS). Although the exact function of TRAIL at the molecular level is unclear, higher levels seem to have protective value. In a recent study of 115 patients with STE-ACS, TRAIL was inversely correlated with the troponin concentration and positively correlated with the left ventricle ejection fraction (LVEF). Thus, low TRAIL levels are associated with a worse LVEF after STE-ACS. These results support the potential role of TRAIL as an independent prognostic factor in patients after ACS [18].

A combination of several biomarkers has been proposed to determine the risk of death in patients with ACS. Stratifying patients according to the number of elevated biomarkers adds prognostic value [19].

## 3. Risk Scores

Clinicians are eager to find patterns that define higher-risk patients to modify treatment strategies. The combination of more than a single risk factor amplifies the probability of MACE.

One of the first scoring systems was proposed by Califf et al. (1988). A total of 5886 consecutively evaluated patients with known significant obstructive coronary disease were stratified according to clinical presentation. A few simple clinical variables (incorporated into the angina score), such as the angina course, angina frequency, and ST-T changes, add independent prognostic value. The angina score alone adds prognostic accuracy to the 2-year survival rate (90% with an angina score of 0 vs. only 68% with an angina score ≥9 in patients with three-vessel disease and a normal ejection fraction) [20].

Subsequently, Braunwald proposed a classification for unstable angina pectoris (UAP) based on the clinical presentation. The categories were subdivided according to the presence or absence of extra-cardiac conditions that could intensify myocardial ischemia, such as infection or hypoxia, or the development of UAP within 2 weeks of myocardial infarction (MI) [21]. Validation on 393 patients with UAP determine the risk of in-hospital cardiac complications. The most prominent risk factor was MI within 14 days, followed by failure to receive a beta-blocker or a rate-lowering calcium-channel blocker before admission, ST-segment depression at admission, and the requirement for intravenous nitroglycerine on admission [22].

The first large study (GUSTO-I) included over 41,000 patients with STE-ACS who were randomized for different thrombolytic strategies (use of streptokinase and subcutaneous heparin; streptokinase and intravenous heparin; accelerated tissue plasminogen activator [t-PA] and intravenous heparin; or a combination of streptokinase plus t-PA with intravenous heparin) defined major risk factors for 30-day mortality. The authors reported that patient age, location of the MI, and physiological characteristics representing myocardial function are the most valid prognostic factors. [23].

The Thrombolysis in Myocardial Ischemia (TIMI) registry trial provided a huge dataset for risk stratification in patients with non-ST elevation (NSTE)-ACS. According to the TIMI-III registry, the authors predicted the risk of MI or death at the 1-year follow-up based on the ECG at admission (1443 patients), which was independently evaluated. The highest rate of MACE was observed in patients who presented with a left bundle branch block (15.8%), followed by ST-segment deviation of a minimum of 0.5 mm. Localized changes in the anterior wall signified a significantly worse prognosis. By contrast, T-wave changes posed the least risk with no association with anatomical localization [24]. The data from the TIMI-III registry trial were also used to determine the risk factors associated with adverse outcomes. The independent predictors of an increase in adverse outcomes are defined (Table 1) [25].

Major progress was achieved by close examination of the TIMI 11B and ESSENCE trials (double-blind placebo-controlled study, unfractionated heparin vs. enoxaparin). The authors used fundamentals to create a risk-scoring system (seven variables) for patients with NSTE-ACS. A progressive and significant increase in the rate of events was observed for each endpoint as the TIMI risk score increased (from 4.7% with 0/1 risk factor to 40.9% for 6/7 risk factors). However, the scoring system indicated a risk of mortality or MI only during the first 14 days [26].

The Global Registry of Acute Coronary Events (GRACE) study was based on robust data collected from 94 hospitals in 14 countries between 1999 and 2002, and more than 17,000 patients were incorporated into the GRACE scoring system. This includes non-selected patients with various acute coronary syndromes. The risk score is based on nine variables. The validity of the GRACE risk score has been demonstrated in various studies. GRACE was initially validated during a 6-month follow-up period. However, subsequently updated GRACE 2.0 with better discrimination and easier usage provides a 3-year follow-up period [28,29,34,35,36].

The PURSUIT score predicts the risk of death or death/MI 30 days after admission, based on data collected in a study that compared eptifibatide (Integrilin) to placebo for managing NSTE-ACS. ACS patients were divided into low, intermediate, and high-risk groups according to the PURSUIT score. [29].

Progressively evolving treatment strategies in patients with ACS show a need for a new scoring system. Percutaneous coronary intervention (PCI) is the gold standard treatment for patients with STE-ACS [37,38,39,40], as well as those with NSTE-ACS. The first of its kind is the PAMI scoring system, which evaluates the prognosis of patients with STE-ACS treated exclusively by PCI with a 1-year follow-up period. Among others (Killip class >1, heart rate >100 beats/min, diabetes, anterior MI, or left bundle branch block), the most significant predictor is age >75 years. Although the authors pointed out the major benefit of its use during an early presentation, the lack of information derived from the catheterization laboratory may be its major limitation [30].

## 4. Anatomical Features and Revascularization

The first detailed scoring system to evaluate the significance of stenotic lesions was proposed by Gensini in 1983. A reduction in the diameter of the lesion was associated with a Gensini score (1, 2, 4, 8, 16, or 32) and multiplied by a coefficient according to the clinical significance [41]. This uncomplicated score was recently compared with the GRACE risk score in the ACS setting. The authors pointed out the benefit of the GRACE risk score for predicting severe coronary lesions. Three-vessel or left main coronary artery disease was present in almost 30% of patients in the high-risk group according to the GRACE risk score compared to only 15.5% in the low-risk group [42].

### 4.1. TIMI Flow

The conditions under which revascularization is performed and the results are crucial for subsequent patient outcome after ACS. Stone et al. describe the importance of baseline TIMI flow. Patients with MI with TIMI-3 flow before revascularization have lower in-hospital rates of new-onset heart failure and hypotension, a lower rate of intubation, and are associated with better initial left ventricular function. The study included 2507 patients and the importance of baseline TIMI flow was confirmed when corrected to post-procedural TIMI-3 flow. There was a 0% mortality rate at the 6-month follow-up in the group with TIMI flow 3 at baseline and after reperfusion, compared to 3.6% mortality in patients with lower initial TIMI flow [43].

Similar results were presented by De Luca et al. in a population of 1791 patients with STE-ACS, divided into high and low risk based on the TIMI risk score. Pre-procedural TIMI flow is significantly related to 1-year mortality in high-risk patients, compared to low-risk patients (16.3% and 3.8% respectively) [44].

Interestingly, pre-procedural TIMI flow was not an independent predictor of 1-year mortality in 3582 patients with NSTE-ACS enrolled in the ACUITY trial [45].

The impact of post-procedural TIMI flow in patients with ACS investigates the COREA-AMI registry on 5025 patients. STE-ACS patients with suboptimal post-procedural flow had a significantly higher mortality rate at the 5-year follow-up than patients with optimal angioplasty results (33.1% vs. 19.6%, respectively). By contrast, TIMI flow after PCI did not present a significant difference in mortality of NSTE-ACS patients [46].

A large prospective registry of 10,000 patients with ACS examine the relationship between pre- and post-procedural TIMI flow and patient outcome. Ndrepepa et al. reported that post-procedural TIMI flow is an independent predictor of 1-year mortality. Although baseline TIMI flow affected early mortality (30-day), it did not influence subsequent mortality from 1 month to 1 year [47]. Post-PCI TIMI flow grade 2 results in intermediate mortality between patients with TIMI flow 0–1 and 3. These results are inconsistent with 2767 patients with NSTE-ACS (Polish ACS registry), where mortality of patients with TIMI 2 is comparable to that of TIMI 0–1 after PCI [48].

A PCI evaluation is an important factor in the Zwolle risk score, which includes almost 1800 STE-ACS patients treated exclusively by PCI. Interestingly, many of the patients (73%) had very low mortality (0.1% at 2 days and 0.5% at 30 days) and a low malignant arrhythmia risk (0.2% risk VT/VF at 48 h), which presents a promising result for selecting patients suited for early discharge [31].

Similarly, three-vessel disease and post-procedural TIMI flow are essential parts of the CADILLAC risk score, presented in 2005. Simple stratification predicts 12-month outcome in patients with acute MI treated with PCI (>2000 patients), with baseline LVEF as the single most powerful predictive variable. [32].

### 4.2. Timing of Revascularization

Early revascularization (TIMACS study) in high-risk NSTE-ACS patients (primary endpoints defined as death, non-fatal MI, or stroke) has no benefit. Patients were selected based on the GRACE scoring system (threshold 140 points). However, the composite outcome of death, MI, refractory ischemia, and shorter hospitalization time was significantly lower. Therefore, an early invasive strategy is recommended in patients with at least one high-risk factor [49]. However, all the patients in this study were pre-treated with an ADP receptor antagonist. An investigation into the effect of pre-treatment in patients with ACS (ACCOAST) reported no anti-ischemic benefit, whereas it elevated the bleeding risk and the risk of delaying coronary artery bypass graft surgery if indicated [50].

These results were confirmed by a meta-analysis conducted by Bellemain-Appaix et al. [51]. Therefore, the current ESC guidelines do not recommend routine pre-treatment of ACS patients with ADP inhibitors.

The EARLY trial compared early vs. delayed intervention in NSTE-ACS patients without pre-treatment. Intermediate or high-risk patients were included and randomized into early (up to 2 h) or delayed (12–72 h) intervention groups. The early invasive strategy in higher-risk patients shows a significant reduction in recurrent ischemic events, but no difference in cardiovascular death [52].

### 4.3. Complexity of Revascularization

An upgrade of the current scoring systems (GRACE and TIMI) is needed, as patients with NSTE-ACS are routinely treated with PCI. The new ACUITY-PCI score combines clinical, laboratory/electrocardiographic, and three angiography findings in NSTE-ACS patients who undergo PCI and predicts 1-year risk of mortality and MI. Subsequent comparison with the above-mentioned and the SYNTAX score in a different population showed promising results with good discrimination and calibration [33].

The complex anatomical features of coronary vessels are included in the SYNTAX score. The primary objective was a detailed assessment of coronary vasculature. Several studies have demonstrated the usefulness of successfully predicting the risk of major ischemic events in patients with stable coronary artery disease or multi-vessel disease. [53,54].

Analysis of the population in the previously mentioned ACUITY trial using the SYNTAX score predicts 1-year clinical outcomes. Palmieri et al. first reported its possible use in patients with NSTE-ACS by demonstrating the accuracy of the score as an independent predictor of 1-year adverse ischemic events [55]. Combining clinical features, such as age, ejection fraction, and renal function (so-called modified ACEF score) with the SYNTAX score resulted in more accuracy and complexity. The clinical SYNTAX score was introduced in 2010 to stratify patients with complex coronary artery disease [56]. Validation in a single-center prospective observational study resulted in significant improvement in the risk prediction for 2-year cardiac death, compared to the original SYNTAX score [57].

The COMPLETE trial showed that patients with multi-vessel disease benefit from staged PCI of non-culprit vessels by lowering the risk of death from a cardiovascular cause or new MI during a 3-year follow-up. No significant difference was observed between staged PCI of the non-culprit vessel during the index hospitalization or several weeks after discharge [58]. The results of a recent meta-analysis confirmed the accuracy of such a treatment [59].

Only the culprit lesion should be treated immediately in patients with MI who present with cardiogenic shock. According to the CULPRIT-SHOCK trial, additional ad-hoc PCI of non-culprit lesions resulted in a higher risk of all-cause death at the 30-day and 1-year follow-ups [60]. Routine use of intra-aortic balloon pumps in such patients does not reduce the mortality rate [61].

### 4.4. Residual Stenosis

Optical coherence tomography (OCT) may quantify anatomical interventional results. Prati et al. retrospectively evaluated the OCT results after urgent/emergent PCI of 507 patients with ACS. Suboptimal OCT stent implantation defined as in-stent minimal lumen area <4.5 mm^2^, dissection >200 µm at the distal stent edge, and reference lumen area < 4.5 mm^2^ in the presence of residual significant plaque at the stent edges. The presence of at least one of the aforementioned shows worse outcomes (17.9% vs. 4.8% cardiac death, target vessel MI, and target lesion revascularization) during the follow-up period (median 345 days). Residual intra-stent plaque/thrombus protrusion (500 µm thickness cut-off) was an independent predictor for a three-fold higher risk of a worse outcome (as previously defined) [62].

Another interesting study published in the *Journal of the American College of Cardiology* describes residual plaque burden in patients with ACS after successful PCI.

Examination by intravascular ultrasound on almost 700 patients identifies 33% of the patients with at least one lesion with a plaque burden >70%. These patients had a two-fold increase in 3-year MACE compared to patients without significant lesions. Importantly, more than two thirds of future clinical events in patients with a significant plaque burden were associated with non-culprit lesions [63].

## 5. Bleeding Risk Score and Others

Treatment with dual antiplatelet therapy (DAPT) produces a higher bleeding risk. The PRECISE-DAPT score was proposed as a tool to predict bleeding risk in patients taking DAPT. Evaluating almost 15,000 patients revealed the efficacy of shortening DAPT (<12 months) in higher-risk patients (PRECISE-DAPT >25). Prolonged DAPT is not associated with an anti-ischemic benefit in these patients. By contrast, low bleeding risk patients (PRECISE-DAPT <25) may profit from prolonging DAPT (>12 months) without additional bleeding risk. Therefore, the PRECISE-DAPT score may select high-risk ACS patients who would benefit from shortening DAPT (if high bleeding risk is present) [64]. The use of the PRECISE-DAPT score is recommended by the ESC guidelines (Class IIb, Level A).

The additional value of the DAPT and PRECISE-DAPT scores is the ability to predict the severity of coronary atherosclerosis. Both scoring systems are independently and positively associated with the Gensini score (mentioned earlier) and the risk of three-vessel disease [65].

The HAS-BLED scoring system can predict hospital and post-discharge bleeding risk and in-hospital mortality on 702 ACS patients (unselected, STE-ACS, and NSTE-ACS) The patients were subdivided into three groups according to the HAS-BLED score (<30, 30–60, and 60–90). A higher score predicts an increased risk of bleeding or in-hospital mortality [66].

Several attempts have been made to evaluate well-established scoring systems. Although the CHA2DS and CHA2DS2-VASc scores are primarily used in the setting of atrial fibrillation, there is a possible application in ACS patients [67]. The CHA2DS2-VASc score is an independent predictor of subsequent MACE in patients with all types of ACS. The very high-risk group has a 3.8-fold increased risk of MACE than the low-risk group. The CHA2DS2-VASc score is comparable to well-established risk scores (GRACE and TIMI) [68].

## 6. Discussion

Attempts to define the probability of MACE in patients with ACS date back to the 1980s when the angina score was presented. Many scoring systems have subsequently emerged, influenced by former therapeutic standards (Table 1). However, in the last 40 years, there has been major progress in management and therapy for ACS (Figure 1).

Among the many scoring systems, the strongest data support the TIMI and GRACE systems, although patients with ACS are treated almost exclusively (TIMI) and mostly (GRACE, PCI in 34%) by thrombolysis, and standard MI treatment was not administered to the patients (47% received statins, 55% ACE-inhibitors and 71% betablockers, aspirin was administered to over 90% after discharge) [75]. Recent comparison questioned the role of GRACE as a universal scoring system. When compared to TIMI, PAMI, and CADILLAC in diabetic patients with STEMI, CADILLAC was superior at predicting 6-month, 1-year, and 2-year cardiovascular outcomes. [76].

Although the use of the most powerful and data-supported scores (GRACE and TIMI) has been recommended for many years, the latest ESC guidelines downgrade the importance of estimating the prognosis based on risk scores (Class I Level B in 2015 vs. Class IIa Level C in 2020), suggesting less interest in scoring systems in upcoming years.

However, due to new therapeutic approaches, it may be beneficial to estimate the prognosis. Differentiating patients with higher hospital/30-day mortality rates may help focus clinicians on the need for intensive care or advanced mechanical circulatory support modalities. Such a score has been proposed exclusively for patients with cardiogenic shock after acute MI by Poss et al. [77].

Carefully selecting very low-risk patients may be beneficial for early convalescence and the return to normal life. The benefits of decreasing hospital time on patient psychological status are significant. Shortening hospitalization time may not only save healthcare resources but also ease pressure on healthcare workers during crucial times, such as during the COVID-19 pandemic.

Despite the vast amount of research, scoring systems are not often used in clinical practice. We hypothesized that time and complex features are the main reasons.

Based on our observations, an optimal scoring system should include a combination of clinical, echocardiographic, and angiographic findings. An easy-to-obtain, valuable clinical trait is Killip class [23,27,28,30,31,32]. Use of baseline ejection fraction should be obligatory. Its importance is apparent from aforementioned CADILLAC risk score [32]. Results of PCI in form of post-procedural TIMI flow should be obtained, as CADILLAC, ZWOLLE, and COREA-AMI points out [31,32,46]. In our opinion, these findings should be a cornerstone of each new scoring system. We believe simplicity is the key, and further diversification of scoring systems based on subtypes of ACS should be avoided.

We hypothesize a simple, user-friendly score reflecting important clinical and up-to-date therapeutic characteristics would be a very useful tool in everyday practice.

## Figures and Tables

**Figure 1 jcm-10-04574-f001:**
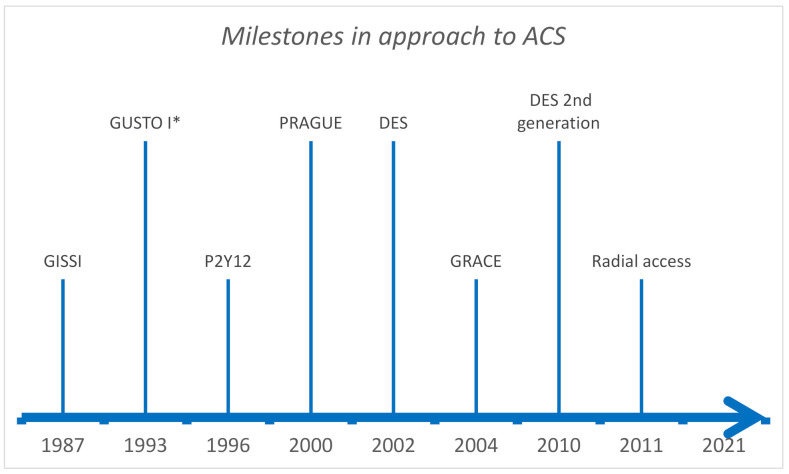
GISSI trial—fibrinolysis with streptokinase [69], GUSTO-I trial—fibrinolysis with alteplase (* same year benefit of PCI vs. streptokinase, 142 patients) [23,37], P2Y12—effectiveness of DAPT (Ticlopidine + Acetylsalicylic acid) over anticoagulation after PCI [70], PRAGUE study—benefit of transferring STE-ACS patient to PCI center [71], DES—drug eluting stents approved in Europe (1-year later in the USA) [72], GRACE—validated GRACE risk score [35], DES 2nd generation—higher efficacy everolimus eluting stent [73], Radial access—safer with less vascular complications vs. femoral access [74].

**Table 1 jcm-10-04574-t001:** Summary of risk models predicting the outcome of patients with acute coronary syndrome (ACS).

Scores to Predict Outcome in Patients with ACS	Year	Type of ACS	Patients	Features	Follow-Up
Angina score [20]	1988	UAP	5886 with symptomatic (75%) stenosis (2946 in the training and 2940 in the validation samples)	Angina course Angina frequency ST-T changes	10 years
Braunwald classification of unstable angina [22]	1989	UAP	393 were prospectively validated	New onset or worsening of angina without rest pain Angina at rest within the last month but not within the last 48 h Angina at rest within 48 h	In-hospital
GUSTO-I trial [23]	1993	NSTE-ACS and STE-ACS	41,021 with myocardial infarction—international randomized trial	Age Systolic blood pressure Killip class Heart rate Location of infarction Previous infarction Age by Killip class interaction Height Time to treatment Diabetes Weight Smoking Choice of thrombolytic therapy Previous bypass surgery	30 days
TIMI-III registry [25]	1997	NSTE-ACS	3318	Age >75 years Heart rate Left bundle branch block Concomitant illness Prior MI or CAD	1 year
TIMI risk score [26] (TIMI 11B and ESSENCE trial)	2000	NSTE-ACS	3910 (TIMI 11B) and 3171 (ESSENCE)	Age >75 years At least 3 risk factors for CAD (family history of CAD, hypertension, hypercholesterolemia, diabetes, current smoking) Significant coronary stenosis ≥50%	14 days
				ST deviations (>0.5 mm) Anginal symptoms (≥2 anginal events in the last 24 h) Use of aspirin in last days Elevated serum Cardiac markers (CK MB and/or Troponin)	
TIMI risk score [27]	2000	STE-ACS	15,078 and 3687 (validation cohort)	Age Diabetes mellitus/hypertension or angina Systolic blood pressure (<100 mmHg) Heart rate (>100) Killip class (II-IV) Weight (<67 kg) Anterior ST elevation or left bundle branch block Time to treatment (>4 h)	30 days
GRACE risk score [28]	2004	NSTE-ACS and STE-ACS	15,007 (developmental cohort) and 7638 (validation cohort)	Age per 10-year increase History of myocardial infarction History of congestive heart failure (Killip class) Systolic blood pressure per 20 mmHg decrease Initial serum creatinine level per 1 mg/dl increase Initial cardiac enzyme elevation ST-segment depression No in-hospital PCI	6 months
PURSUIT trial [29]	2000	NSTE-ACS	9461 (eptifibatide vs. placebo)	Age Sex Worst CCS class in previous 6 weeks Signs of heart failure ST depressions on presenting ECG	30 days
PAMI risk score [30]	2004	STE-ACS	3252 treated by PCI	Age Killip class Heart rate Diabetes Anterior myocardial infarction or left bundle branch block	1 year
ZWOLLE risk score [31]	2004	STE-ACS	1791 treated by PCI, 747 (validation cohort)	Age Killip class Ischemia time (>4 h) Anterior infarction Three-vessel disease TIMI flow post	30 days
CADILLAC risk score [32]	2005	NSTE-ACS and STE-ACS	2082 and 900 (validation cohort)	Baseline left ventricular ejection fraction Renal insufficiency Killip class Age (>65) Anaemia Three-vessel disease Final TIMI flow	1 year
ACUITY-PCI risk score [33]	2012	NSTE-ACS	1692 and 846 (validation cohort)	Extent of coronary disease Small vessel/diffuse coronary artery disease Bifurcation lesion present Baseline cardiac biomarker elevation or ST-segment deviation Insulin-treated diabetes Renal insufficiency	1 year
GRACE risk score 2.0 [34]	2013	NSTE-ACS and STE-ACS	32,037 (development cohort) and 2959 (validation cohort)	Age per 10-year incrementSystolic blood pressure per 20 mmHg increment Pulse per 30 beats/min increment Cardiac arrest at admission Positive initial biomarkers ST deviation Renal insufficiency Use of diuretics in first 24 h after presentation	3 years

UAP, unstable angina pectoris; STE-ACS, ST elevation acute coronary syndrome; NSTE-ACS, non-ST elevation acute coronary syndrome; ST-T, ST segment and T wave electrocardiogram changes; MI,myocardial infarction; CAD, coronary artery disease; CK-MB, creatine kinase -myoglobin binding; PCI, percutaneous coronary intervention; CCS, Canadian Cardiovascular Society grading of angina pectoris; ECG, electrocardiogram.

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
