# Peer review of "Risk Stratification of Patients with Acute Coronary Syndrome"

_jcm, 2021, doi:10.3390/jcm10194574_

Round 1
Reviewer 1 Report
This is an excellent review and update on the subject.
Author Response
Dear Reviewer,
We would like to thank you for Your positive comments, we highly appreciate your feedback.
Sincerely,
David B.
Reviewer 2 Report
Dear authors,
this review is a short and simply readable summary of such contributions in the field of risk stratification of ACS.
Studies cited are to-date cornerstones in daily ACS risk stratification.
However, some minor revision should be assessed:
1) English style and format must be revised, especially in the introduction, as sentences are quiet complex and misleading. Therefore, the introduction paragraph must be completely rewritten. Moreover, passive forms throughout the review are extensively used and must be limited.
2) Abstract is simply a copy of introduction paragraph. Please rewrite originally
3) Table 1 is probably too much full of information. Please limit to most commonly used scores and please separate each score by a line as text alignment is not correct.
4) Table 2 is quiet unrelated to text. You should consider to introduce a table with anatomic risk scores.
5) Discussion paragraph is too brief. Please add more specifics to each observation.
6) Conclusions are quiet obvious. Please add some personal suggestions to such suggested user friendly score (which parameters should be assessed, which biomarkers, etc.)
Author Response
Dear Reviewer,
First of all, thank you for your valuable comments. We are confident that Your suggestions increased quality of the paper.
- Completly rewritten introduction, made revision of english style throughts the whole text.
- We made correction in table 1. so information are clear and separated.
- There are changes in discussion. We add personal suggestion what future scoring system should look like and why.
Thank you,
Sincerely David B.